# Frequency of an X-Linked Maternal Variant of the Bovine *FOXP3* Gene Associated with Infertility in Different Cattle Breeds: A Pilot Study

**DOI:** 10.3390/ani12081044

**Published:** 2022-04-17

**Authors:** Md Shafiqul Islam, Mitsuhiro Takagi, Keun-Woo Lee, Hye-Sook Chang, Hiroaki Okawa, Muchammad Yunus, Tita Damayanti Lestari, Martia Rani Tacharina, Shahnaj Pervin, Tofazzal Md Rakib, Akira Yabuki, Osamu Yamato

**Affiliations:** 1Laboratory of Clinical Pathology, Joint Faculty of Veterinary Medicine, Kagoshima University, 1-21-24 Korimoto, Kagoshima 890-0065, Japan; si.mamun@ymail.com (M.S.I.); bestofme@daum.net (H.-S.C.); s.pervin30@yahoo.com (S.P.); rakibtofazzal367@gmail.com (T.M.R.); yabu@vet.kagoshima-u.ac.jp (A.Y.); 2Department of Pathology and Parasitology, Faculty of Veterinary Medicine, Chattogram Veterinary and Animal Sciences University, Khulshi, Chattogram 4225, Bangladesh; 3Laboratory of Theriogenology, Joint Faculty of Veterinary Medicine, Yamaguchi University, 1677-1 Yoshida, Yamaguchi 753-8511, Japan; mtakagi@yamaguchi-u.ac.jp; 4Department of Veterinary Internal Medicine, College of Veterinary Medicine, Kyungpook National University, 80, Daehak-ro, Buk-gu, Daegu 41566, North Gyeongsang, Korea; kwolee@knu.ac.kr; 5Animal and Plant Quarantine Agency, 177, Hyeoksin 8-ro, Gimcheon-si 39660, Gyeongsangbuk-do, Korea; 6Guardian Co., Ltd., 2794-127 Nishi-Beppu-cho, Kagoshima 890-0033, Japan; okawa0117@guardian-vet.com; 7Faculty of Veterinary Medicine, Airlangga University, Campus C, Mulyorejo, Surabaya 60115, Indonesia; muchammad-y@fkh.unair.ac.id (M.Y.); titadlestari@fkh.unair.ac.id (T.D.L.); martia.rt@fkh.unair.ac.id (M.R.T.)

**Keywords:** bovine *FOXP3* gene, regulatory T cell, allele frequency, Japanese black, Holstein Friesian, Hanwoo, Madura, cow, infertility

## Abstract

**Simple Summary:**

Recently, an X-linked maternal single-nucleotide polymorphism (SNP) in the upstream of the bovine *FOXP3* gene (NC_037357.1: g.87298881A>G, rs135720414) was identified in Japanese Black (JB) cows in association with recurrent infertility. The objective of this study was to evaluate the frequency of this SNP in different cow breeds. Between 2018 and 2021, a total of 809 DNA samples were obtained from 581 JB, 73 Holstein Friesian (HF), 125 Korean Hanwoo (KH), and 30 Indonesian Madura (IM) cows, which were genotyped based on real-time polymerase chain reaction analysis. The G allele frequency was found to be relatively high in local IM (0.700), moderate in dairy HF (0.466), and low in beef JB (0.250) and KH (0.112) cows, with differences in the frequencies between each group being statistically significant (*p* < 0.005) as per the Fisher’s exact test. The results obtained in this study indicate that the G allele frequencies of the identified *FOXP3* gene SNP differ markedly in different breeds of taurine and indicine cattle.

**Abstract:**

Immune adaptation plays an essential role in determining pregnancy, which has been shown to be dependent on sufficient immunological tolerance mediated by FOXP3^+^ regulatory T cells. Recently, an X-linked maternal single-nucleotide polymorphism (SNP), located 2175 base pairs upstream of the start codon in the bovine *FOXP3* gene (NC_037357.1: g.87298881A>G, rs135720414), was identified in Japanese Black (JB: *Bos taurus*) cows in association with recurrent infertility. However, with the exception of JB cows, the frequency of this SNP has yet to be studied in other cow populations. In this study, we thus aimed to evaluate the frequency of this SNP in different cow breeds. Between 2018 and 2021, a total of 809 DNA samples were obtained from 581 JB, 73 Holstein Friesian (HF: *B. taurus*), 125 Korean Hanwoo (KH: *B. taurus coreanae*), and 30 Indonesian Madura (IM: a crossbreed between *B. indicus* and *B. javanicus*) cows, which were genotyped using a TaqMan probe-based real-time polymerase chain reaction assay designed in this study. The frequency of the G allele was found to be relatively high in local IM (0.700), moderate in dairy HF (0.466), and low in beef JB (0.250) and KH (0.112) cows, with differences in the frequencies between each group being shown to be statistically significant (*p* < 0.005) using Fisher’s exact test. The results obtained in this study indicate that the G allele frequencies of the identified the SNP differ markedly in different breeds of taurine and indicine cattle. Given these findings, it would thus be important to evaluate the relationships between high frequencies of the G allele and infertility in different breeds.

## 1. Introduction

Reproductive failure and infertility in cattle are of major concern in the dairy and beef industries worldwide. They are associated with the complex interactions among genetic, physiological, environmental, and managerial factors [1]. These factors contribute to problems with conception, inter-calving period, and delivery of healthy calves; postpartum complications, reduced milk yield; and assortative mating [1,2]. The fertility in lactating dairy cows is currently declining worldwide, with repeat breeding being identified as one of the most important reproductive problems affecting fertility [3].

Maternal factors known to contribute to infertility include age, oocyte defects, endocrine dysfunction, nutritional and hormonal abnormalities, genital tract infections, and genetic alterations, although it is often difficult to identify the predominant cause when different precipitating factors coexist [1]. In recent years, the roles of maternal immune regulatory cells and their regulatory genes that may also contribute to infertility have gained increasing attention in both human and animal studies. In particular, the development, differentiation, and immune suppressive functions of regulatory T (Treg) cells have been a prime focus of numerous studies following the identification of *FOXP3*, a master gene that encodes a transcription factor regulating the development and function of these Treg cells [4]. Recent studies in human medicine have also demonstrated the vital role played in maternal–fetal immunity in maintaining a normal pregnancy, which is associated with the induction of different immunocompetent cells. For example, dysfunctional CD4^+^ CD25^+^ Treg cells have been found to be linked to implantation failure [4,5].

In 2001, a mutation in the *FOXP3* gene of scurfy mice was identified as a new marker for the suppressive effect of Treg cells [6], and since then, several mutations in this gene have been identified in human patients, particularly in males suffering from immune dysregulation, polyendocrinopathy, enteropathy, and X-linked (IPEX) syndrome [7], as well as autoimmune diseases and preeclampsia [8]. The effect of these mutations has been proposed to negatively affect the differentiation of maternal Treg cells, which could in turn promote fetus-specific effector T-cell activation and subsequent infertility [9]. Numerous studies have also detected an association between unexplained infertility or pregnancy complications and reduced endometrial *FOXP3* mRNA expression in women [10], and the findings of a recent human study have also indicated that a mutation in the *FOXP3* promoter region is associated with recurrent spontaneous abortions in Chinese Han populations [11].

Contrastingly, despite the association between *FOXP3* gene variants and reproductive pathology, comparatively few studies have examined *FOXP3* gene-related defects in veterinary medicine. In a recent genome-wide association study (GWAS), however, an X-linked maternal single-nucleotide polymorphism (SNP), located 2175 base pairs upstream of the start codon in the bovine *FOXP3* gene (NC_037357.1: g.87298881A>G, rs135720414), was identified in Japanese Black (JB: *Bos taurus*) cows in association with recurrent infertility [9]. Reporter assays indicated that this SNP G allele was associated with reduced levels of *FOXP3* transcription, which may be a maternal immunogenic factor underlying the identified association between recurrent infertility in repeat breeding cows and higher frequencies of the G allele [9]. Consequently, it is reasonable to speculate that this SNP could serve as a valuable target in efforts to enhance the fertility of cow herds. 

To the best of our knowledge, with the exception of JB cows, the SNP has yet to be surveyed in populations of other bovine breeds. Accordingly, in this study, we specifically sought to analyze the frequencies of this SNP in JB, Holstein Friesian (HF: *B. taurus*), Korean Hanwoo (KH: *B. taurus coreanae*), and Indonesian Madura (IM: a crossbreed between *B. indicus* and *B. javanicus*) cow populations. 

## 2. Materials and Methods

The experiments conducted in this study were performed in accordance with the guidelines regulating animal use and ethics at Kagoshima University (no. VM15041; approval date: 29 September 2015) and Yamaguchi University, Japan (no. 40, 1995; approval date: 27 March 2017), and oral informed consent was obtained from cooperating farmers.

### 2.1. Sample Collection and DNA Extraction

From 2018 to 2021, blood samples were collected from 809 clinically healthy cows (female): 581 JB, 73 HF, 125 KH, and 30 IM cows. The JB cows were born and raised on several private JB cattle farms in Kagoshima and Soo cities, Kagoshima Prefecture, Japan. The HF cows were derived from several commercial dairy herds in Fukuoka Prefecture, Japan. The KH cows were born and raised on several farms near Taegu City in Korea, and the IM cows were born and raised on Madura Island, Indonesia. The blood samples (no more than 1 mL) were obtained by jugular or caudal venipuncture and were spotted onto Flinders Technology Associates filter papers (FTA card; Whatman International Ltd., Piscataway, NJ, USA) and stored in a refrigerator (4 °C) until use for the extraction of DNA. DNA was extracted from discs punched out of the blood-impregnated FTA cards following appropriate treatment, as previously described [12].

### 2.2. Genotyping of the SNP

The primers and TaqMan minor groove binder (MGB) probes used for the real-time polymerase chain reaction (RT-PCR) assays (the sequence of which are listed in Table 1) were designed based on the sequence of bovine *FOXP3* (NCBI Reference Sequence NC_037357.1). These primers and probes, each of which was linked to a fluorescent reporter dye (6-carboxyrhodamine or 6-carboxyfluorescein) at the 5′-end and a non-fluorescent quencher dye at the 3′-end, were synthesized by a commercial company (Applied Biosystems, Foster City, CA, USA). RT-PCR amplifications were carried out in a final volume of 5 µL consisting of 2× PCR master mix (TaqMan GTXpress Master Mix; Applied Biosystems), 80× genotyping assay mix (TaqMan SNP Genotyping Assays; Applied Biosystems) containing the specific primers, TaqMan MGB probes, and template DNA. A negative control containing nuclease-free water instead of template DNA was included in each run. The cycling conditions consisted of 20 s at 95 °C, followed by 50 cycles of 3 s at 95°C and 20 s at 60 °C, with a subsequent holding stage at 25 °C for 30 s. The data obtained were analyzed using StepOne version 2.3 (Applied Biosystems). Several DNA samples with three different genotypes (A/A, A/G, and G/G) from four different bovine breeds were used to validate the genotyping assay, following genotype confirmation based on Sanger sequencing (Kazusa Genome Technologies Ltd., Kisarazu, Japan). The sequence around the SNP (NC_037357.1: g.87298881A>G, rs135720414) was confirmed based on the publicly available bovine genome sequence (ARS-UCD1.2).

### 2.3. Statistical Analysis

Differences in allele frequencies between each group were statistically analyzed using Fisher’s exact test, with *p* values of less than 0.05 considered to indicate a statistically significant difference.

## 3. Results

In this study, we developed an RT-PCR assay using TaqMan MGB probes, which enables us to clearly identify all possible genotype combinations (A/A, A/G, and G/G). The genotyping results were found to be consistent with the Sanger sequencing results (Figure 1).

The results of our survey of the four assessed cattle breeds are shown in Table 2. The A and G alleles were found to be present in all breeds examined in this study, with the frequency of the G allele being relatively high in IM (0.700), moderate in HF (0.466), and low in JB (0.250) and KH (0.112) cows. Collectively, the overall G allele frequency of the study cohort was 0.265. HF, JB, and KH cows were characterized by a predominant A allele at this locus, whereas in IM cows, the G allele was detected more frequently than the A allele. Moreover, differences in the allele frequencies between different groups were found to be statistically significant (*p* < 0.005) using Fisher’s exact test. 

## 4. Discussion

In a previous GWAS analysis, Arishima et al. identified an SNP (NC_037357.1: g.87298881A>G, rs135720414) in the upstream of the *FOXP3* gene of JB cattle. This SNP was associated with a reduction in *FOXP3* transcription, which in turn was linked to a reduction in the number of maternal Treg cells and led to infertility or repeat breeding [9]. Screening for this allele would ideally necessitate a simple reliable genotyping assay. In the present study, we accordingly designed a TaqMan MGB probe-based RT-PCR assay, the use of which provided clear-cut genotyping results (A/A, A/G, and G/G) for the non-risk-type A and risk-type G alleles of the SNP. Moreover, by using FTA cards for sampling on cattle farms, we were able to eliminate the need for traditional multi-step DNA extraction and purification procedures, and this, combined with a relatively short amplification time (less than 1 h), facilitated the rapid genotyping and screening of the target SNP in less than 2 h, which is comparable with genotyping surveys performed for a range of bovine, canine, and feline genetic diseases [12,13,14]. 

Our survey of different cattle breeds using the newly designed assay revealed that the risk of carrying the G allele was approximately 0.265, calculated from 458 A/A, 273 A/G, and 78 G/G genotypes detected in a population of 809 cows, and thus notably less likely than harboring the non-risk A allele (Table 2). The frequency of the G allele detected in JB cows in the present study (0.250) is similar to that (0.23) in the JB population screened by Arishima et al. [9]. Among the other assessed breeds, we found that the frequency of the G allele in KH cows (determined for the first time in this study) was significantly lower (0.112) than that in JB cows. Both JB and KH are beef cattle, which may have originally possessed the same maternal genetic factors associated with infertility and have unintentionally been selected with respect to the A allele. However, infertility and the problems associated with miscarriage/stillborns have yet to be sufficiently resolved in these beef breeds [15,16,17,18,19], thereby indicating that there remain other maternal factors responsible for these problems. Furthermore, the KH breed is believed to be a hybrid of taurine (*B. taurus*) and indicine (*B. indicus*) cattle [20,21], and it is thus conceivable that the significant difference in G allele frequency between JB and KH cows may be attributable to the blood from indicine cattle.

Compared with that of the other three assessed breeds, the risk of carrying the G allele in HF dairy cows proved to be moderate (0.466). It would thus be particularly beneficial to focus on the SNP with regard to reducing the levels of infertility in these dairy cows. We are currently undertaking a follow-up survey, in which we are examining the subsequent reproductive performance of each of the HF cow screened in the present study, with a view toward clarifying the associations between the genotype and fertility.

The IM breed of cattle is one of the Indonesian-native bovine breeds developed by crossing zebu (*B. indicus*) and banteng (*B. javanicus*), the herds of which are predominantly reared by small-scale farmers on Madura Island, East Java, not only for beef production, but also as working draught animals and for the purposes of racing and show [22,23]. In the present study, we established that compared with cows of the other three breeds, these cattle have a notably high likelihood of carrying the G allele (0.700), and we speculate that this high G allele frequency could be attributable to the genetic contribution of banteng cattle. However, this remains to be confirmed, given that frequencies of the G allele in these cattle have yet to be surveyed. Furthermore, it is speculated that the persistence of this detrimental allele in populations could be attributable to a lack of appropriate breeding management, as IM cattle are generally bred to maintain specific traits by natural mating (82.7%) with local IM breeding bulls [19]. In this regard, there is also a high risk of inbreeding and thus the propagation of single mutant genes within the cattle herds, owing to the comparatively small number of IM bulls that are available for mating within the limited area in which they are reared [22]. These conditions are conducive to considerable reductions in genetic variation, which may increase the prevalence of specific alleles such as the SNP G allele. Among the major reproductive problems associated with IM cattle is repeat breeding [24], which is conceivably linked to the high frequency of the G allele. If this is the case, the SNP would be a potentially valuable genetic marker.

In order to maximize the utility of the SNP as a target in a range of cattle breeds, particularly the HF an IM breeds, further studies will be necessary to evaluate the reproductive performance and biochemical parameters, such as metabolic profiles, among different breeds. 

## 5. Conclusions

On the basis of the results obtained in this study, we established that current frequencies of the detrimental G allele of the SNP in the upstream of the bovine *FOXP3* gene are particularly high in local populations of IM cows compared with those of three other assessed breeds, among which, G allele frequencies are low in KH and JB cows, and moderate in HF dairy cows. Given these findings, it would thus be important to evaluate the relationships between high frequencies of the G allele and infertility in different breeds. Given that the moderate to high G allele frequencies in IM and HF cows are plausibly associated with the infertility of these breeds, it would be interesting to evaluate the relationship between the high frequency of the risk-type G allele and infertility. In this regard, the genotyping assay developed in this study could make a notable contribution to surveying the bovine populations.

## Figures and Tables

**Figure 1 animals-12-01044-f001:**
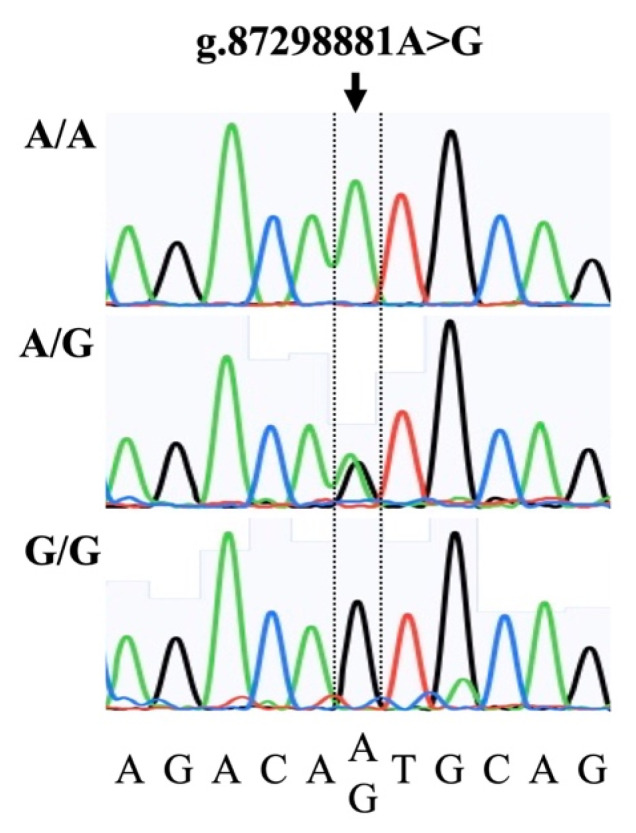
Representative Sanger sequencing electropherograms illustrating the A/A, A/G, and G/G genotypes associated with a single-nucleotide polymorphism (arrow; g.87298881A>G) in the upstream of the bovine *FOXB3* gene.

**Table 1 animals-12-01044-t001:** Sequences of the primers and probes used in the real-time polymerase chain reaction (RT-PCR) assay and Sanger sequencing for a single-nucleotide polymorphism in the upstream of the bovine *FOXP3* gene.

Primer/Probe	Sequence 5′ to 3′ (mer)	Reporter (5′)	Quencher (3′)	Tm (°C)	Concentration (nM)
RT-PCR:					
Forward primer	CCATGTGGCTTCTGAGAAATAGTCA (25)	NA	NA	67.1	450
Reverse primer	TACCTGGAGGGCCAGACT (18)	NA	NA	62.3	450
Probe for the A allele	TCTTCCTGCATTGTCTG (17)	VIC	NFQ	50.0	100
Probe for the G allele	TCTTCCTGCACTGTCTG (17)	FAM	NFQ	52.0	100
Sanger sequencing:					
Forward primer	AGGGCTCAGATGCAGAC (17)	NA	NA	54.0	NA
Reverse primer	GGATATGGTCTGTCTGGT (17)	NA	NA	54.3	NA

Tm, melting temperature calculated using Oligo Calculator (http://www.ngrl.co.jp/tools/0217oligocalc.htm (accessed on 16 April 2022); NA, not applicable; VIC, 6-carboxyrhodamine; FAM, 6-carboxyfluorescein; NAQ, non-fluorescent quencher. The underlined letter in the sequence of the probe for the G allele indicates the corresponding adenine to a guanine transition (NC_037357.1: g.87298881A>G, rs135720414) in the upstream of the bovine *FOXB3* gene.

**Table 2 animals-12-01044-t002:** The number of cattle genotyped and the frequencies for the bovine *FOXP3* single-nucleotide polymorphism in different cattle breeds.

Cattle Breed	Number of Examined Cows	Number of A/A Allele (%)	Number of A/G Allele (%)	Number of G/G Allele (%)	G Allele Frequency
Japanese Black	581	333 (57.3)	205 (35.3)	43 (7.4)	0.250 *
Holstein Friesian	73	24 (32.9)	30 (41.1)	19 (26.0)	0.466 *
Korean Hanwoo	125	98 (78.4)	26 (20.8)	1 (0.8)	0.112 *
Indonesian Madura	30	3 (10.0)	12 (40. 0)	15 (50. 0)	0.700 *
Total	809	458 (56.6)	273 (33.7)	78 (9.6)	0.265

***** Differences in frequencies between different groups were statistically significant (*p* < 0.005) using Fisher’s exact test.

## Data Availability

Not applicable.

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
