# Peer review of "Frequency of an X-Linked Maternal Variant of the Bovine FOXP3 Gene Associated with Infertility in Different Cattle Breeds: A Pilot Study"

_animals, 2022, doi:10.3390/ani12081044_

Round 1
Reviewer 1 Report
In this manuscript authors provide some descriptive data on frequency of the X-linked maternal variant of the bovine FOXP3 gene which they claim to be associated with infertility in cattle. They have used four different breeds of cattle and samples collected from different localities.
Major concern:
In my opinion, data provided do not support the conclusions of the study. This is particularly evident from the fact that nothing much is elucidated regarding this SNP and its implications in these cattle breeds except for the Japanese breed. Similarly, I have noticed that the discussion chapter itself is a mere piling of extant literature and at some instances the paragraphs are poorly connected to topic of current study.
Introduction is very poorly written and must be optimized. There are many redundant paragraphs and poor connections between sentences. Ideally, authors should include some information about reproductive performances of all breeds covered in this study. The very objective of this study is poorly described and perhaps buried in the long unnecessary paragraphs.
In material and methods some of important information is missing. For instance, age and sex, parity, and history of animals in relation to reproductive health of all animals studied must be described.
Information of different localities must be disclosed: For instance, author report that “The JB and HF cows were born and raised in different areas in Japan”. Authors need to provide information (including environment, housing etc.) about those different areas and disclose their names.
Authors need to provide information, both in methods and results, whether they validated these sequences (data reported in the present study) using already available datasets or clearly state if the datasets are not available publicly?
As I mentioned before, the discussion chapter must be overhauled, and the conclusion toned down.
Other General Comments:
Given that line numbers are missing in the document, I was unable to mark specific line corrections. Apologies for that. However, I have listed some of the major weaknesses below.
English language and syntax are very poor and must be edited extensively.
Data provided is not enough to merit its publication as a full-length research article, instead it should be considered for short communication type.
Title is so long and confusing and somehow misleading because no functional analysis was made in this study. Lease make necessary corrections.
Both summary and abstract sections need to be complete overhauled. The purpose of simple summary should be kept in mind while writing this section. Abstract is simple repetition of literature and should be revised extensively.
Reviewer 2 Report
The article focuses on an undoubtedly critical topic and the experimental methodology appears adequate.
However, describing the different frequencies of the G allele in different races without contextualizing them seems to be an end in itself. What I mean, it is unclear whether or not breeds with a higher G allelic frequency have better reproductive performance. For example, a simple literature search aimed at pointing out the phenotypic differences in these breeds (listing tangible data, e.g. days open interval) could support what is reported under discussion. Furthermore, an extensive review should be made especially in the introduction describing in a more linear way the reasons behind this study.
In addition, line numbers are missing from the pdf manuscript
Introduction:
- Please split this sentence “They are associated with the complex interactions among genetic, physiological, environmental, and managerial factors that contribute to problems with conception, inter-calving period, and delivery of healthy calves; postpartum complications, reduced milk yield; and repeat breeding. [1, 2]”
- Could you substitute "repeat breeding" with assortative mating
- The sentence “The fertility in cows is dependent on a range of contributory factors, including the quality of bull semen, appropriate insemination during estrus, and certain environmental factors, such as stress and season. However, most cases in which the causal factors remain unidentified are believed to be associated with maternal factors that differ among individual cows” contains the same information as the sentence reported 1. Please rewrite the first part of the introduction following a logical order, otherwise, it is too difficult to follow
- I did not have the point of this paragraphs: “Maternal factors known to contribute to infertility include age, oocyte defects, endocrine dysfunction, nutritional and hormonal abnormalities, genital tract infections, and genetic alterations, although it is often difficult to identify the predominant cause when different precipitating factors coexist [1]. Genetic disorders and detrimental traits linked to inherent gene and chromosomal abnormalities also compromise reproductive efficiency in different breeds of cattle, including Holstein Friesian (HF: taurus) and Japanese Black (JB: B. taurus) cows in Japan [5, 6]. In addition, different maternal genetic factors, including the altered expression of oocyte mitochondrial DNA and apoptotic genes and changes in the endometrial gene expression at different stages of the ovarian cycle, have been established as causal factors contributing infertility in dairy cows [7, 8].” do you mean that there are maternal causes that have a genetic basis? If it could have been written more concisely without all these lists
Discussion
- Change “which might could be a maternal immunogenetic factor underlying the identified association between recurrent infertility in repeat breeding cows and higher frequecies of the G allele [14].” In “which may be a maternal immunogenic factor underlying the identified association between recurrent infertility in repeat breeding cows and higher frequencies of the G allele”
- “On the basis of these observations, it is reasonable to assume that this SNP could serve as a valuable marker, the identification of which enable the reduction and/or elimination of animals with inferior fertility” It is a too hard statement, both in the cited study and in this one, the influence of this gene is not tangibly demonstrated (i.e look the percentage of genetic variance explained by that snp)
- “However, we cannot exclude the possibility that the observed differences in allele frequency may have been unintentionally biased by the preferential use of cows with better fertility related to the A allele.”
- “Poor fertility in dairy cows has become a serious problem worldwide with substantial economic losses due to increases in additional inseminations, repeat breeding, veterinary treatment, management, and involuntary culling [1]. Intensive genetic selection for high milk yields during the past five decades has been linked to a progressive erosion of the genetic potential for fertility in dairy breeds [27]. A number of maternal genetic factors, along with the condition of bull semen and environmental factors, can potentially contribute to dairy cow infertility, especially among cows subjected to repeated breeding, with an incidence ranging from 9% to 24% of lactating cows [8], and accordingly, this may account for the significantly higher G allele frequency in dairy cows than in JB and KH beef cows.” Please remove this part
Conclusion:
- Please Replace “particular value to evaluate” with” interesting to evaluated”
Reviewer 3 Report
This paper provides a simple and straightforward analysis of allele frequencies in four cattle breeds for a SNP previously found to be associated with repeat rebreeding (a proxy for infertility) in Japanese Black cattle. The SNP in question appears to be a sequence variant, and the authors designed a protocol using direct Sanger sequencing of the region to determine SNP genotype. The importance of FOXP3, the gene 2 kb downstream from the SNP, on fertility made up a majority of the discussion, despite the fact that nothing in the author’s analysis was able to directly make these connections.
General comments:
The authors make conclusions that are not supported by their analyses. For example, the discussion on the first paragraph of page 6 is centred on the notion that this RT-PCR assay should be used for selection of one allele over the other. While I agree that this SNP may be a valuable addition to large marker panels used for genetic selection, with no evidence of the effect of this SNP in breeds other than JB, the results of this analysis do not support that it should be used for single SNP selection. The authors need to clarify their conclusions throughout the discussion.
The single analysis used in this study would be greatly strengthened if the results could be validated using publicly available sequence datasets. I would not expect rare breeds such as IM to have much if any publicly available data. However, there are publicly available HF and KH sequence sets, and I expect JB may as well. Repositories such as NCBI SRA could be mined to accomplish this.
Throughout the document, including the simple summary, the SNP in question is referred to as being “in” the FOXP3 gene. This is incorrect and needs to be corrected in every instance.
Arishima et al., the paper that this paper’s analysis was based on, went into great detail about LD patterns linking this sequence variant with FOXP3. The paper would greatly benefit by the addition of discussion regarding LD in the region surrounding the tested variant. Especially because this variant does not appear to be functional.
What are the fertility rates of KH and IM cattle? Could the differences in fertility rates between the four breeds of cattle help explain differences in allele frequency?
Specific comments:
In multiple instances through the text p-values are denoted with a lower case “p”, ie: p < 0.05. In this context the correct notation is with a capital and italicized “P”, ie: P < 0.05. Please change throughout, and take note of where this was incorrectly used in the table/figure captions.
The abbreviation SNP was not defined at first use. This needs to be defined in the abstract as well.
In the methods you state that the primers used were based off of NCBI Ref Seq NC_037057.1. This is the ARS-UCD1.2 reference assembly. However, the SNP position listed (SNP: g.92,377,936A>G) is from UMD3.1. Please update the SNP position to the ARS assembly, and in each instance that it is referenced, please include the ref seq number corresponding to the chromosome assembly. (Ex: NC_037341.1:g.23338890G>T)
Please add the dbSNP name for the variant in question (rs or ss).
Please add a reference number for the animal use and ethics approval utilized by this project.
Were all the animals sampled the same sex? If not, were there differences in allele frequencies by sex, within breed?
Round 2
Reviewer 1 Report
Although some of my comments were not addressed or partly addressed, but the revised version is much better than before.
English language still needs attention.
Reviewer 3 Report
No further comments or suggestions.